# Wnt/β-Catenin Pathway Is Involved in Cadmium-Induced Inhibition of Osteoblast Differentiation of Bone Marrow Mesenchymal Stem Cells

**DOI:** 10.3390/ijms20061519

**Published:** 2019-03-26

**Authors:** Lu Wu, Qinzhi Wei, Yingjian Lv, Junchao Xue, Bo Zhang, Qian Sun, Tian Xiao, Rui Huang, Ping Wang, Xiangyu Dai, Haibo Xia, Junjie Li, Xingfen Yang, Qizhan Liu

**Affiliations:** 1Center for Global Health, School of Public Health, Nanjing Medical University, Nanjing 211166, China; wuluxxl@126.com (L.W.); chaos_key@163.com (J.X.); lattaqueen@gmail.com (Q.S.); m15720801737@163.com (T.X.); dxy19930726@163.com (X.D.); haiboxia346@163.com (H.X.); m15895903083@163.com (J.L.); 2The Key Laboratory of Modern Toxicology, Ministry of Education, School of Public Health, Nanjing Medical University, Nanjing 211166, China; 3Food Safety and Health Research Center, Guangdong Provincial Key Laboratory of Tropical Disease Research, School of Public Health, Southern Medical University, Guangzhou 510515, China; cnwei99@163.com (Q.W.); cnzhangbo@126.com (B.Z.); 4Guangdong Provincial Center for Disease Control and Prevention, Guangzhou 511430, China; luiassay@sina.cn (Y.L.); huangr@gdiph.org.cn (R.H.); wangp@gdiph.org.cn (P.W.)

**Keywords:** osteogenesis, bone marrow mesenchymal stem cells, cadmium, Wnt/β-catenin pathway

## Abstract

Cadmium is a common environmental pollutant that causes bone damage. However, the effects of cadmium on the osteogenic differentiation of bone marrow mesenchymal stem cells (BMMSCs) and its mechanism of action in this process are unclear. Here, we determined the effects of cadmium chloride (CdCl_2_) on the osteogenic differentiation of BMMSCs and the potential mechanism involved in this process. As determined in the present investigation, CdCl_2_, in a concentration-dependent manner, affected the viability of BMMSCs and their cytoskeletons. Exposure to 0.1 or 0.2 µM CdCl_2_ inhibited osteogenic differentiation of BMMSCs, which was reflected in the down-regulation of osteoblast-related genes (*ALP*, *OCN*, *Runx2*, *OSX*, and *OPN*); in suppression of the protein expression of alkaline phosphatase (ALP) and runt-related transcription factor 2 (Runx2); and in decreased ALP activity and capacity for mineralization. Moreover, mRNA microarray was performed to determine the roles of these factors in BMMSCs treated with CdCl_2_ in comparison to control BMMSCs. As determined with the microarrays, the Wingless-type (Wnt), mothers against decapentaplegic and the C. elegans gene Sam (SMAD), and Janus kinase-Signal Transducers and Activators of Transcription (JAK-STAT) signaling pathways were involved in the effects caused by CdCl_2_. Moreover, during differentiation, the protein levels of Wnt3a, β-catenin, lymphoid enhancer factor 1 (LEF1), and T-cell factor 1 (TCF1) were reduced by CdCl_2_. The current research shows that CdCl_2_ suppresses the osteogenesis of BMMSCs via inhibiting the Wnt/β-catenin pathway. The results establish a previously unknown mechanism for bone injury induced by CdCl_2_.

## 1. Introduction

Cadmium, a heavy metal and persistent environmental toxicant, has a biological half-life of 10–30 years [1,2]. It is released into the environment through natural processes and human activities, such as mining and smelting, which may lead to contamination of soils and to increased cadmium uptake by crops and vegetables that are grown for human consumption [3]. A cluster of adverse health effects, including bone damage and nephrotoxicity, occur with long-term exposure to water and food contaminated with cadmium. Exposure to high levels to cadmium reduces bone mineral density (BMD) manifested as osteoporosis and osteomalacia [4]. In addition, population-based studies from several countries reveal an association between decreased BMD and low-level environmental exposure to Cd [3]. The molecular and genetic basis by which CdCl_2_ causes diseases, however, remains largely unknown.

As a rigid organ, bone is in a dynamic balance and is self-renewed by the activities of osteoblasts and osteoclasts [5]. Osteoclasts resorb bone and then recruit bone marrow mesenchymal stem cells (BMMSCs) for subsequent differentiation and bone formation, which is responsible for continuous bone remodeling [6]. BMMSCs, as potential progenitor cells regulated by transcription factors, differentiate into a variety of cell lineages including osteoblasts, chondrocytes, and adipocytes [7,8]. Based on the properties mentioned above, MSCs play a vital role in tissue repair and regeneration and have great potential applications in tissue engineering and regenerative medicine [9,10,11,12,13]. Compared to immortalized cell lines, which have abnormal characteristics after transformation, using BMMSCs to assess the toxicity of cadmium has many advantages [5]. Therefore, we choose BMMSCs to investigate the toxicity of CdCl_2_ on bone formation.

Various cytokines, growth factors, and signaling pathways, including the Wnt/β-catenin pathway, are essential in the regulation of osteoblast differentiation [14]. There has many research reporting that Wnt/β-catenin pathway is associated with differentiation of BMSCs. Recently, a research revealed that the inhibition of Wnt pathway suppresses osteogenic differentiation of BMSCs during osteoporosis [15]. In BMSCs, insulin-like growth factor binding protein 7 (IGFBP7) regulates the osteogenic differentiation via Wnt/β-catenin pathway [16]. The activation of canonical Wnt signaling leads to cytoplasmic stabilization and increased nuclear translocation of β-catenin, which activates T-cell factor/lymphoid enhancer factor (TCF/LEF) transcription factors and switches from repression to activation of TCF/LEF target genes [17]. To date, there has been no report on the relationship between the Wnt pathway and the inhibition of osteogenic differentiation of BMMSCs induced by CdCl_2_. Knowledge of the regulatory Wnt pathway and its role in the osteogenic differentiation of BMMSCs is limited.

The present study investigated the inhibition of osteogenesis in BMMSCs by CdCl_2_. The expression patterns of mRNAs in BMMSCs and CdCl_2_-treated BMMSCs (CdCl_2_-BMMSCs) were analyzed. For CdCl_2_-BMMSCs, changes of signaling pathways that may be involved in the suppression of osteogenic differentiation of BMMSCs induced by CdCl_2_ were addressed. Further, during osteogenic differentiation, CdCl_2_ inhibited the Wnt pathway. The results provide an understanding of bone dysfunction induced by CdCl_2_.

## 2. Results

### 2.1. Purity and Potential to Differentiate of BMMSCs

Primary BMMSCs were isolated from rat bone marrow and passaged. Cells adhered to the plastic culture dishes and demonstrated a typical small spindle-shaped morphology (Figure 1A). BMMSCs have positive and negative markers, including Sca-1, CD44, CD73, CD90, CD105, and CD45 [18,19,20]. Confirming the identity of BMMSCs, flow cytometry analysis showed that the cells had high levels of CD44 (99.80 ± 0.19%) and CD90 (96.54 ± 0.46%) but limited expression of the hematopoietic marker CD45 (1.93 ± 0.56%) (Figure 1B). The data confirm that the isolated cells were positive for mesenchymal-associated markers. Multipotency is another criterion for defining BMMSCs [21]. After 21 days of adipogenic induction, numerous lipid droplets were detected by staining with Oil Red O (Figure 1C). After 14 days of osteogenic induction, numerous mineralized nodules were evident, as revealed by alizarin red S staining (Figure 1D). These results indicated that the primary BMMSCs cultured had adequate purity and a high potential to differentiate.

### 2.2. Low Concentrations of Cadmium Have No Effect on Cell Viability, but High Concentrations of Cadmium Reduce Cell Viability of BMMSCs in a Concentration-Dependent Manner

To evaluate CdCl_2_ cytotoxicity, BMMSCs were incubated with 0, 0.25, 0.5, 1.0, 2.0, 4.0, 6.0, 8.0, 10, or 20 μM CdCl_2_ for 24 h. For low concentrations (≤1.0 μM), there were no significant differences in cell viability between the CdCl_2_-treated and control groups (Figure 2A). However, high concentrations (≥2.0 μM) of CdCl_2_ reduced viability of BMMSCs in a concentration-dependent manner. Since we evaluated the effects CdCl_2_ on osteogenic differentiation for 7 days or more, we also assessed cell viability after incubation with various concentrations of CdCl_2_ for 7 days. BMMSCs were incubated with 0, 0.1, 0.2, 0.5, 1.0, 2.0, 3.0, 4.0, or 5.0 μM CdCl_2_ for 7 days. Low concentrations (≤0.2 μM) had no effect on viability of BMMSCs, but high concentrations (≥0.5 μM) reduced the viability of BMMSCs in a concentration-dependent manner (Figure 2B). In addition, 5-ethynyl-2′-deoxyuridine (EdU) assay was performed to detect the proliferation rate of BMMSCs treated by CdCl_2_ for 24 h or 7 days_._ We found that the results of EdU assay were consistent with that of 3-(4,5-dimethylthiazol-2-yl) -2,5-diphenyltetrazolium bromide (MTT) (Figure 2C). Therefore, we chose concentrations of 0.1 and 0.2 μM CdCl_2_ to treat BMMSCs in further studies. Furthermore, we treated the cells with 0, 0.1, or 0.2 μM CdCl_2_ for 24 h, after which immunofluorescence and confocal microscope analysis showed that CdCl_2_ changed the morphology and atrophied the skeleton of BMMSCs (Figure 2D). These data indicated that CdCl_2_ had no effect on cell viability but damaged cell morphology of BMMSCs, and that high concentrations inhibited cell viability in a concentration-dependent manner.

### 2.3. Cadmium Suppresses the Osteogenesis of BMMSCs

When BMMSCs seeded into plates reached 60% confluence, CdCl_2_ was supplemented into osteogenic induction medium to determine if CdCl_2_ affected the osteogenesis of BMMSCs. BMMSCs were exposed to 0, 0.1, and 0.2 μM CdCl_2_ for various times. Quantitative Real-Time Polymerase Chain Reaction (qRT-PCR) and RT-PCR revealed that the osteogenic-related mRNA levels of alkaline phosphatase (*ALP*), osteocalcin (*OCN*), runt-related transcription factor 2 (*Runx2*), osterix (*OSX*), and osteopontin (*OPN*) in CdCl_2_-BMMSCs decreased after 3 days and 7 days (Figure 3A–G). ALP activity in cells treated with CdCl_2_ was inhibited in a concentration-dependent manner (Figure 3H). Western blot assays demonstrated that CdCl_2_ suppressed the protein expressions of ALP and Runx2 in BMMSCs (Figure 3I,J). In CdCl_2_-BMMSCs, ALP staining was faint compared to the control after 14 days of osteo-induction (Figure 3K). Alizarin red S staining was used to reveal the effects of CdCl_2_ on the mineralization potential of BMMSCs. Since mineralized nodules are typically observed at terminal differentiation [22], cells were osteo-induced for 14 days. In control BMMSCs, there were numerous dark brown nodules; CdCl_2_ caused lower numbers of relatively smaller nodules (Figure 3K). These data demonstrated that CdCl_2_ suppress the osteogenesis of BMMSCs.

### 2.4. The Effects of Cadmium on mRNA Expression Profiles in BMMSCs

The expression of global genes was evaluated in a gene microarray performed with BMMSCs incubated in osteogenic differentiation medium containing 0.2 μM CdCl_2_. A heat map showed the top 40 down-regulated and the 40 up-regulated mRNAs in CdCl_2_-BMMSCs relative to control BMMSCs and the color indicator means gene expression, the darker the color, the higher the gene expression (Figure 4A). The detailed information of these top regulated mRNAs is presented in Appendix A. mRNAs differentially expressed between control and CdCl_2_-treated cells were identified through scatter plots (Figure 4B). The red dots meant upregulated genes (>2.0-fold change) and the blue dots meant downregulated genes (>2.0-fold change). In CdCl_2_-BMMSCs, 242 mRNAs were down-regulated, and 230 were up-regulated. Gene oncology (GO) analysis showed that pathways related to osteogenic differentiation, including the Wnt signaling pathway, the mothers against decapentaplegic and the C. elegans gene Sam (SMAD) signaling pathway, and the Janus kinase-Signal Transducers and Activators of Transcription (JAK-STAT) signaling pathway, and pathways associated with cell motility were affected by CdCl_2_ (Appendix A). The detailed information of Wnt signaling related differentially mRNAs in BMMSCs treated with CdCl_2_ is presented in Appendix A. Through these pathways, CdCl_2_ may regulate the osteogenic differentiation of BMMSCs.

### 2.5. Cadmium Inhibits the Wnt/β-Catenin Pathway in BMMSCs

The Wnt signaling appears to be involved in regulating the differentiation BMMSCs [23,24]. The activation of Wnt signaling facilitates osteogenic differentiation [25]. Moreover, an mRNA microarray indicated that the Wnt signaling pathway are involved in the inhibition of osteogenic differentiation of BMMSCs induced by CdCl_2_. Therefore, we hypothesized that CdCl_2_ causes a Wnt/β-catenin dysfunction and thus affects osteogenesis. Wnt3a, β-catenin, LEF1, and TCF1 are molecules in the Wnt/β-catenin pathway. Exposure of BMMSCs to CdCl_2_ for 24 h decreased protein levels of these proteins (Figure 5A,B). After BMMSCs were exposed to the osteogenic medium containing Wnt3a, the decreased protein levels of Wnt3a, β-catenin, LEF1, and TCF1 were rescued (Figure 5C,D). In order to exclude the potential effects of the multi-factors in osteogenic medium, we also checked the expression of Wnt3a, β-catenin, LEF1, and TCF1 in BMMSCs treated with CdCl_2_ (without osteogenic medium), which showed that the protein expression levels increased (Figure 5E,F). Thus, in BMMSCs, Wnt/β-catenin pathway is inhibited by CdCl_2_.

### 2.6. Wnt/β-Catenin Pathway is Involved in the Suppression of Osteogenesis Induced by Cadmuim in BMMSCs

To further investigate the role of Wnt/β-catenin pathway in the suppression of osteogenesis induce by CdCl_2_, BMMSCs were exposed to CdCl_2_ at concentrations of 0 or 0.2 μM with or without the treatment of Wnt3a and induced to differentiate into osteoblasts by osteogenic induction medium. After the treatment of Wnt3a for 7 days, Western blots showed that the protein levels of ALP and Runx2 were increased (Figure 6A,B), and the ALP activity was also increased (Figure 6C). After the treatment of Wnt3a for 14 days, the ALP staining showed that the faint staining induced by 0.2 μM CdCl_2_ was rescued, and the alizarin red s staining determined that the smaller and fewer dark-brown colored nodules were increased (Figure 6D). These data demonstrate that, in BMMSCs, Wnt/β-catenin pathway is involved in the inhibition of osteoblast differentiation caused by CdCl_2._


## 3. Discussion

In this work, we found that CdCl_2_ suppressed the osteogenic differentiation of BMMSCs. After treatment of BMMSCs with CdCl_2_, the expression of osteogenic-related genes was reduced, and ALP staining and alizarin red staining confirmed that CdCl_2_ had an adverse effect on the osteogenic differentiation capacity of BMMSCs. In addition, we also performed mRNA microarray to identify biological processes that may be involved in the suppression of the osteogenesis and found that several classical signaling pathways may be involved. We measured the expressions of related molecules involved in the Wnt pathway and found that key molecules in this pathway were low in the presence of CdCl_2_, indicating that activation of the Wnt pathway was inhibited. 

Epidemiological studies have evaluated effects of cadmium among various populations exposed by environmental routes. These studies demonstrate that increased cadmium exposure correlates with decreased bone mineral density (BMD) [26]. For rats, those exposed to cadmium show low BMD and pathological changes in the microstructure of bone, including increased trabecular separation and decreased trabecular connectivity. [27]. Cadmium has both an indirect and direct mode of influencing bone turnover [28]. Cadmium reduces bone formation through inhibition of osteoblast activity and promotes bone resorption, which is related to the impairment of bone cell metabolism [29,30]. However, the specific mechanism for bone damage caused by CdCl_2_ is unclear, especially from the perspective of osteogenic differentiation of BMMSCs. In the present study, BMMSCs from rat bone marrow were used to investigate the effects of CdCl_2_ on the viability and osteogenesis of BMMSCs. 

BMMSCs derived from rats show surface markers whose characteristics indicate that they can differentiate into osteoblasts. The changes of cytoskeleton can affect the osteogenic differentiation of BMMSCs [31]. In our present research, with the treatment of CdCl_2_, the cytoskeleton of BMMSCs were damaged, which indicated that CdCl_2_ may affect the osteoblast differentiation. ALP is an early marker during osteogenic differentiation of BMMSCs, and the level of ALP activity can reflect the degree of osteogenic differentiation [32,33]. Runx2 is involved in osteogenic differentiation and bone formation and is an essential transgenic gene factor [34], it can promote expression of genes during early stages of osteogenic differentiation [35]. OSX is a downstream target associated with the osteogenesis of Runx2 [36]. In addition, OPN and OCN are involved in embryonic bone formation and are activated during bone remodeling [37,38]. In the present study, after osteo-induction for 1, 3, or 7 days with osteogenic media, the mRNA expressions of *ALP*, *Runx2*, *OCN*, *OSX*, and *OPN* were up-regulated in a time-dependent manner. In addition, we selected concentrations of Cd that did not reduce BMMSCs proliferative activity for 7 days as determined by the MTT assay. We used osteogenic media containing various concentrations of Cd for osteogenic induction for 1, 3, and 7 days. After treatment, expressions of the osteogenesis-related genes were lower in the group exposed to CdCl_2_ relative to the control group. Further, the inhibitory effect of CdCl_2_ on osteogenic differentiation of BMMSCs was reflected in lower ALP activity, lower protein levels of ALP and Runx2, and less calcification.

Bone remodeling is an indispensable condition for maintaining bone balance, and bone formation is destroyed in the pathological state of bone. The imbalance between osteoblast-mediated bone formation and osteoclast-mediated bone resorption is an important cause of reduced bone mass and altered bone structure [39]. The formation and calcification of osteoblast plays a crucial role during bone modeling and remodeling, which can be regulated by the Wnt/β-catenin pathway. The results of many investigations about the Wnt pathway in the context of bone generally show increases in canonical signaling through this pathway, leading to increased osteoblastogenesis and bone formation [40]. Most components of the Wnt signaling pathway are functional in BMMSCs, which are often used as a model for bone [41]. Canonical Wnt signaling is involved in self-renewal and in lineage-selective determination of BMMSCs. In humans and in mice, there is a relationship between Wnt signaling and bone mass. Canonical Wnt signaling promotes bone formation by mechanisms including stem cell renewal [42], stimulation of osteoblast proliferation, and induction of osteoblastogenesis [43]. Therefore, we hypothesized that CdCl_2_ damages Wnt signaling and thereby affects osteogenesis.

The canonical Wnt/β-catenin pathway is triggered when the Wnts are combined with Wnt ligands. The binding of Wnt with Frizzled and LRP5/6 initiates to form a complex of proteins, which subsequently recruit the Axin complex. The protein complex, which consists of axin, glycogen synthase kinase 3 (GSK3), and the adenomatosis polyposis coli (APC) protein, stabilizes β-catenin. β-catenin entering into the nucleus activates many target genes, such as the lymphoid enhancer-binding factor/T cell factor (LEF/TCF) transcription factor family, which promotes the expression of many osteoblast-related genes [44,45]. In the present study, we established two groups of cells, normal BMMSCs and CdCl_2_-BMMSCs, and performed mRNA microarray to find altered genes. The bioinformatics analysis indicated that Wnt/β-catenin signaling is involved in the suppression of osteogenesis induced by CdCl_2_. Therefore, we examined expression of proteins involved in the Wnt/β-catenin pathway, including Wnt3a, β-catenin, LEF1, and TCF1. The expressions of these were lower in CdCl_2_-BMMSCs than in controls, and the proteins in the group treated with 0.2 μM Cd were lower than those of the group treated with 0.1 μM CdCl_2_. Activation of the Wnt/β-catenin signaling pathway is related to the osteogenic differentiation of BMMSCs, but, in this study of BMMSCs, the expression of proteins in the Wnt/β-catenin pathway decreased after their treatment with CdCl_2_. CdCl_2_ affected activation of the Wnt/β-catenin pathway. After the treatment of Wnt3a, the protein levels of Wnt3a, β-catenin, LEF1, and TCF1 and the degrees of osteoblast differentiation of BMMSCs were rescued. Thus, inhibition of osteogenic differentiation of BMMSCs is associated with inhibition of activation of the Wnt/β-catenin pathway by CdCl_2_.

## 4. Materials and Methods

### 4.1. Isolation and Culture of Primary BMMSCs

Primary BMMSCs were harvested from 3-week-old Sprague-Dawley (SD) rats as described previously [5]. Briefly, the femurs and tibias of rats were dissected, and bone marrow was flushed out with syringes and centrifuged at 800 rpm for 5 min to collect the cells. Cells were cultured in Modified Eagle medium-Alpha (α-MEM) (Corning Cellgro, New York, NY, USA) supplemented with 10% fetal bovine serum (FBS) (Gibco, USA), 1% L-glutamine (Gibco, Gaithersburg, MD, USA), 1% penicillin-streptomycin (Gibco, USA), and 1% HEPES (Gibco, USA) and were maintained in a humidified 37 °C/5% CO_2_ incubator with changes of medium every other day. When the confluence reached 80–90%, BMMSCs were harvested by treating them with 0.25% trypsin-EDTA (Gibco, USA) and were passaged for further culture. BMMSCs, in passages 3–5, were used for experiments. All of the experiments and procedures involving animals were approved by the Institutional Animal Care and Use Committee (IACUC) of Nanjing Medical University (Identification code:11015; Approval Number: IACUC-1705014; Date: 24th April 2017).

### 4.2. Adipogenic Differentiation

BMMSCs were induced to adipogenic differentiation in Dulbecco’s modified Eagle medium-High glucose (DMEM-HG) (Hyclone, Logan, Utah, USA) supplemented with 10% FBS, 1 μmol/L dexamethasone, 100 μmol/L indomethacin (Sigma Aldrich, St. Louis, MS, USA), 0.5 mmol/L 3-isobutyl-1-methylxanthine (Sigma Aldrich, USA), 5 μg/mL insulin (Sigma Aldrich, USA), 1% l-glucose, 1% penicillin/streptomycin, and 1% HEPES.

### 4.3. Osteogenic Differentiation

BMMSCs were induced to osteogenic differentiation in DMEM-HG medium supplemented with 10% FBS, 10 nmol/L dexamethasone (Sigma Aldrich, USA), 10 mmol/L β-glycerol phosphate (Sigma Aldrich, USA), 50 μg/mL ascorbic acid (Sigma Aldrich, USA), 1% L-glucose, 1% penicillin-streptomycin, and 1% HEPES.

### 4.4. CdCl_2_ Treatment

For (Sigma Aldrich, USA) exposure of BMMSCs during osteogenic differentiation, CdCl_2_, at concentrations of 0, 0.1, or 0.2 μM, was added into the osteogenic induction medium before replacing the growth medium, once BMMSCs in the culture plates reached 60% confluence.

### 4.5. Cell Surface Marker Expression of BMMSCs

Expression of surface markers was assessed by flow cytometry using monoclonal antibodies for CD44 and CD90 (positive BMMSC markers) and CD45 (negative BMMSC marker) (BD Biosciences, San Jose, CA, USA). BMMSCs at passage 3 were suspended in PBS and incubated with each antibody at a concentration of 0.5 μg/mL, along with unstained BMMSCs and isotype controls.

### 4.6. Cell Proliferation Assays

Cell viability was assessed by 3-(4,5-dimethylthiazol-2-yl)-2,5-diphenyltetrazolium bromide (MTT) (Sigma Aldrich, USA) following the manufacturer’s instructions. BMMSCs (1 × 10^4^ cells per well) were seeded in 96-well plates. 24 h later, BMMSCs were exposed to 0, 0.25, 0.5, 1.0, 2.0, 4.0, 6.0, 8.0, 10, 20 μM or to 0, 0.1, 0.2, 0.5, 1.0, 2.0, 3.0, 4.0, 5.0 μM CdCl_2_ for 24 h or for 7 days. After treatment, MTT reagent, diluted to a concentration of 0.5 mg/mL with osteogenic-inducing medium, was added to the plates, and they were incubated at 37 °C for 4 h. Next, the MTT was replaced by dimethyl sulfoxide (Sigma Aldrich, USA), and preparations were incubated at 37 °C for another 15 min, and then the plates were agitated softly for 15 min. The absorbance of each well was recorded at 490 nm by an Infinite M200 Pro instrument (TECAN, Switzerland). Each assay was repeated at least three times independently.

For the 5-ethynyl-2′-deoxyuridine (EdU) proliferation assay, treated BMMSCs cells were seeded into 96-well plates for EdU assay to assess the proliferation ability, and each well contained 2 × 10^4^ cells. Then completed medium containing of 0.1% EdU A solution (RiboBio, Guangzhou, China) were added into 96-well after cells were attached to the well, and then incubated for two hours in an incubator. Subsequent experiments were carried out in strict accordance with the manufacturer’s instructions. Finally, photos were taken by fluorescence microscope.

### 4.7. Confocal Microscopic Analysis

Cells (7 × 10^4^ /well) were cultured on coverslips (In Vitro Scientific) and treated with 0, 0.1, or 0.2 μM CdCl_2_ for 24 h, after which cells were fixed with 4% paraformaldehyde for 30 min. After treating of cells with 0.1% Triton X-100, their cytoskeletons were stained with anti-α-tubulin antibody (1:200, Beyotime, China) or anti-F actin antibody (1:200, Abcam, Cambridge, MA, USA), together with respective fluorescence secondary antibodies (Alexa Flour 488 and Alexa Flour 546) (1:200, Beyotime, Shanghai, China). Cell nuclei were stained with 4′,6-diamidino-2-phenylindole (DAPI) (1:1000, Beyotime, Shanghai, China) for 5 min. Images were visualized and captured with a Nikon E800 confocal microscope (Nikon, Tokyo, Japan).

### 4.8. Oil Red O Staining

Cells (7 × 10^4^ /well) were seeded in 24-well plates, and adipogenesis was induced for 14 days. Cells were then fixed with 4% formalin solution for 15 min and incubated with 0.5% Oil Red O (Sigma Aldrich, USA) for 40 min. After washing the plates with 60% isopropanol, images were made at 10× magnification using an optical microscope (Nikon, Japan).

### 4.9. Alkaline Phosphatase (ALP) Assay

ALP activity was determined with Sensolyte® pNPP Alkaline Phosphatase Assay Kits (Anaspec, USA) according to manufacturer’s instructions. BMMSCs were seeded in 96-well plates at a density of 1 × 10^4^ cells/well. At 24 h after plating, the cells were exposed to 0, 0.1, or 0.2 μM CdCl_2_ with or without the treatment of Wnt3a, and osteogenic differentiation was induced as described above. Cells were washed twice with assay buffer, lysed with Triton-X-100, and collected in microcentrifuge tubes. After incubation at 4 °C for 10 min under agitation, cells were centrifuged at 2500× *g* for 10 min to collect the supernatant. The supernatant was incubated with P-nitrophenyl phosphate (pNPP) substrate solution, and the absorbance was read at 405 nm with an Infinite M200Pro (TECAN, Switzerland). The ALP activity was normalized against protein concentration measured with BCA Protein Assay Kits (Beyotime Institute of Biotechnology, China).

### 4.10. Alkaline Phosphatase (ALP) Staining

Leukocyte Alkaline Phosphatase Kits (Sigma Aldrich, USA) were used for ALP staining according to the manufacturer’s instructions. BMMSCs were seeded in 24-well plates at a density of 7 × 10^4^ cells/well in growth culture medium. Until the confluence reached 60%, BMMSCs were exposed to CdCl_2_ at concentrations of 0, 0.1, or 0.2 μM with or without the treatment of Wnt3a and induced to differentiate into osteoblasts by osteogenic induction medium. After 10 d of induction, the cells were fixed with 4% formaldehyde and 5% citrate in acetone at room temperature for 30 s. The fixed cells were washed with PBS and incubated with 0.2% naphthol AS-BI and 0.2% diazonium salt at room temperature for another 15 min. After washing the plates with PBS, images were taken at 10× magnification under an optical microscope (Nikon, Japan).

### 4.11. Alizarin Red S (ARS) Staining

BMMSCs were seeded in 24-well plates at a density of 7 × 10^4^ cells/well and were exposed to CdCl_2_ at concentrations of 0, 0.1, or 0.2 μM with or without the treatment of Wnt3a, and induced to differentiate into osteoblasts by osteogenic induction medium for 14 days. The cells were then washed with PBS, fixed with 10% formaldehyde at room temperature for 10 min, and incubated with 40 mM alizarin red S (Sigma Aldrich, USA) solution at room temperature for 20 min. After discarding the working solutions and washing the plates with PBS 4 times, images were made at 10× magnification under an optical microscope (Nikon, Japan).

### 4.12. RNA Preparation and Quantitative Real-Time Polymerase Chain Reaction (qRT-PCR)

Total RNA was isolated from cells with TRIzol reagent (Invitrogen Life Technologies Co, Carlsbad, CA, USA) according to manufacturer’s protocol. The purity and concentration of total RNA was assessed with a NanoDrop 2000 (Thermo Fisher Scientific, Waltham, MA, USA). Reverse transcription was accomplished using Prime Script™ RT Reagent Kits with gDNA Eraser (Perfect Real Time, Takara, Kusatsu, Japan) with 1 μg of RNA according to the manufacturer’s instructions. qRT-PCR was performed with an ABI7900 Fast Real-Time System (Applied Bio systems, Waltham, MA, USA) using SYBR Premix Ex Taq™ Kits (Takara, Japan). GAPDH was used as an internal standard, and the relative expressions of genes were calculated by the 2^−ΔΔCt^ method [46]. Primers sequences are shown in Table 1.

### 4.13. Reverse-Transcriptase Polymerase Chain Reaction (RT-PCR)

Total RNA (1 μg) was treated with 10 U of RNase R (Epicentre Technologies Corp., Madison, WI) in 1× RNase R reaction buffer in a total volume of 10 μL. The mixture was incubated at 37 °C for 1 h. Total RNA (1 μg) was transcribed into cDNA by HiScript II Q RT Supermix (Vazyme Biotech). The PCR reactions were evaluated by checking the PCR products on 2% *w*/*v* agarose gels. The intensities of bands were quantified with ImageJ software.

### 4.14. Western Blots

Cell were lysed with RIPA (Beyotime Institute of Biotechnology, China) as described previously [5], and protein concentrations were quantified with BCA Protein Assay kits (Beyotime Institute of Biotechnology, China). Equal amounts (50 μg) of protein were separated by 10% sodium dodecyl sulfate-polyacrylamide gel electrophoresis and were transferred to polyvinylidene fluoride membranes (Millipore, Billerica, MA, USA). Membranes were incubated overnight at 4 °C with a 1:1000 dilution of anti-glyceraldehyde 3-phosphate dehydrogenase (anti-GAPDH, Beyotime) and an antibody for ALP (Abcam, USA), Runx2 (Cell Signaling Technology, Boston, MA, USA), Wnt3a (Abcam, USA), β-catenin (Abcom, USA), LEF1(Cell Signaling Technology, USA), or TCF1 (Cell Signaling Technology, USA). After additional incubation with a 1:1000 dilution of HRP-conjugated goat anti-mouse and goat anti-rabbit secondary antibodies (Jackson ImmunoResearch, USA) for 1 h, the immune complexes were detected by enhanced chemiluminescence (Cell Signaling Technology, USA). The intensities of bands were quantified with ImageJ software.

### 4.15. Microarray for Analysis of Gene Expression

BMMSCs were cultured with CdCl_2_ at a concentration of 0.2 μM in osteogenic differentiation medium and each group included one sample that mixed by three parallel samples. Afterwards, cells were lysed, and total RNA was extracted for analysis of whole genome expression. Osteogenic differentiation medium without Cd was used as a control. The RNA samples were labeled and hybridized with 1.65 μg of Cy3-labeled cRNA using Gene Expression Hybridization Kits (Santa Clara, USA) in a Hybridization Oven (Santa Clara, USA), according to the manufacturer’s instructions. After 17 h of hybridization, slides were washed in staining dishes (Thermo Shandon, US) with Gene Expression Wash Buffer Kits (Santa Clara, US). Samples were scanned with an Agilent Microarray Scanner (Santa Clara, US) with the default settings: Dye channel: Green, Scan resolution = 3 μm, PMT 100%, 20 bit. Data were extracted with Feature Extraction software 10.7 (Santa Clara, CA, USA). Specific representations of heatmap and scatter plot were analyzed by bioinformatics using “pheatmap” package. The gene oncology was analyzed by bioinformatics using “ggplot2” package.

### 4.16. Statistical Analyses

All data values were expressed as means ± standard deviations (SD). Graphpad 7.0 was applied for statistical analyses. One-way analysis of variance (ANOVA) was used for comparisons of means among multiple groups, and a multiple-range least significant difference (LSD) was used for inter-group comparisons. All statistical analyses were performed with SPSS 19.2, and the data were marked with (*) for *p* < 0.05, or (**) for *p* < 0.01.

## 5. Conclusions

In summary, our investigations show that CdCl_2_ suppresses the osteogenic differentiation of BMMSCs and address the effects of CdCl_2_ on mRNA microarray. In this process, CdCl_2_ caused decreases of Wnt3a, β-catenin, LEF1, and TCF1, which are involved in the Wnt pathway, which regulates differentiation of BMMSCs. Thus, in BMMSCs exposed to CdCl_2_, the Wnt/β-catenin pathway may regulate inhibition of osteogenic differentiation (Figure 7). Our findings establish a mechanism for CdCl_2_-induced bone damage and have public health implications for developing strategies to reduce CdCl_2_ exposure and thereby to mitigate its harmful effects.

## Figures and Tables

**Figure 1 ijms-20-01519-f001:**
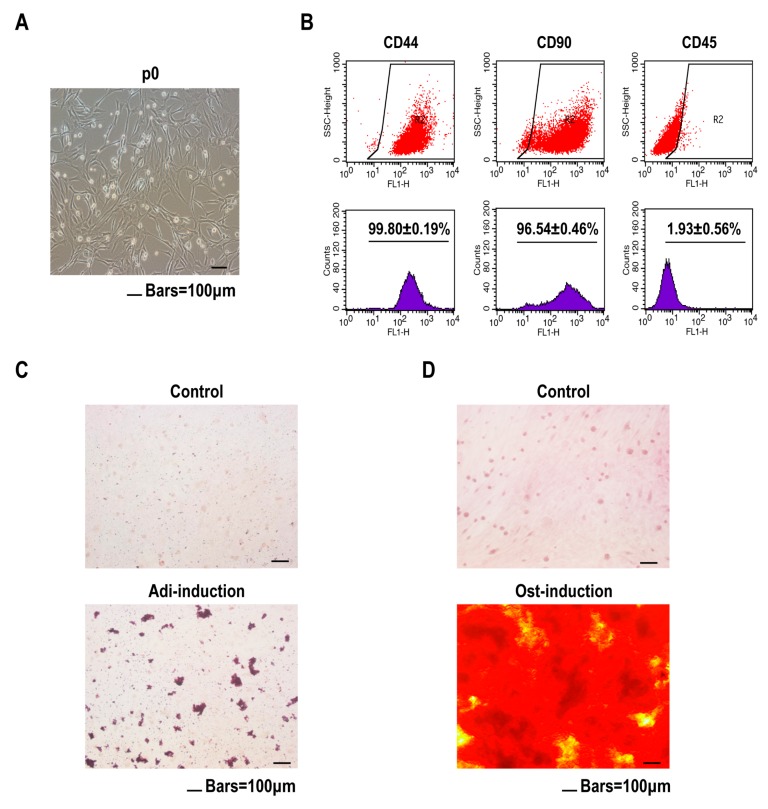
Prepared bone marrow mesenchymal stem cells (BMMSCs) have good purity and a high potential to differentiate. BMMSCs were cultured for 4 days. (**A**) Cell morphology of BMMSCs was shown under a light mirror. (**B**) The cell surface markers of BMMSCs were determined by flow cytometry analysis (mean ± SD, *n* = 3). BMMSCs were treated for adipogenic induction for 21 days. (**C**) Lipid droplets were determined by Oil Red O staining showed. BMMSCs were treated for osteogenic induction for 14 days. Scale bar, 100 μm. (**D**) Mineralization nodules were determined by alizarin red S staining. Scale bar, 100 μm.

**Figure 2 ijms-20-01519-f002:**
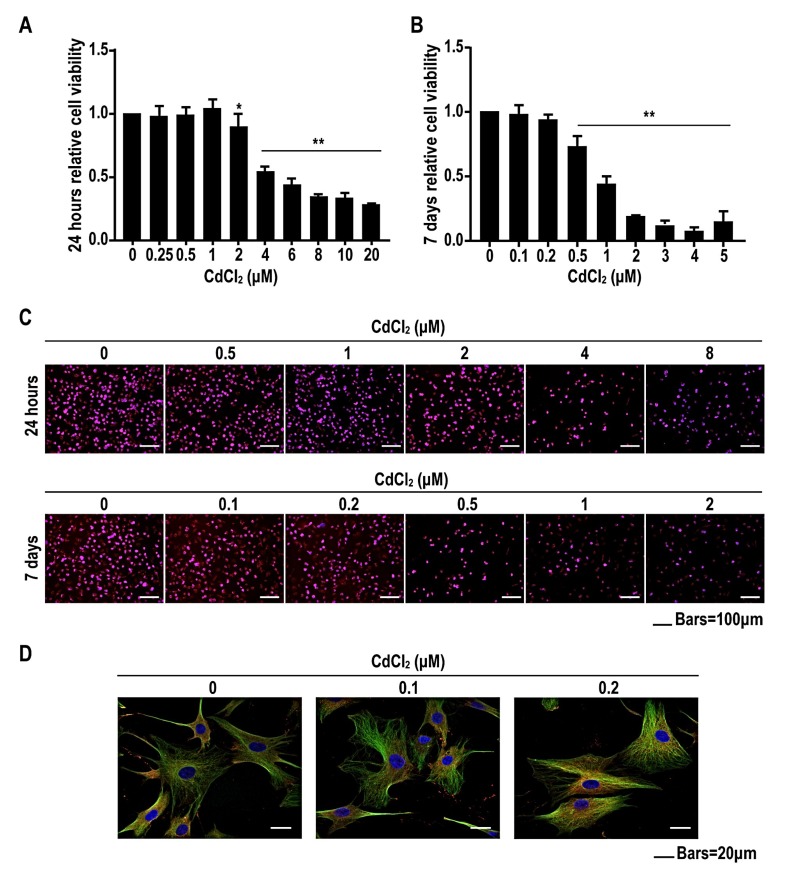
Low concentrations of cadmium have no effect on cell viability, and high concentrations of cadmium reduce cell viability of BMMSCs. BMMSCs were exposed to 0, 0.25, 0.5, 1.0, 2.0, 4.0, 6.0, 8.0, 10, or 20 μM CdCl_2_ for 24 h. (**A**) The viability of BMMSCs was detected by 3-(4,5-dimethylthiazol-2-yl) -2,5-diphenyltetrazolium bromide (MTT) analysis (mean ± SD, *n* = 5). * *p* < 0.05, ** *p* < 0.01, different from control BMMSCs. BMMSCs were treated with 0, 0.1, 0.2, 0.5, 1.0, 2.0, 3.0, 4.0, or 5.0 μM CdCl_2_ for 7 days. (**B**) The viability of BMMSCs was detected by MTT analysis. (mean ± SD, *n* = 5). ** *p* < 0.01, different from control BMMSCs. BMMSCs were exposed to 0, 0.5, 1, 2, 4, and 8 μM CdCl_2_ for 24 h, or 0, 0.1, 0.2, 0.5, 1, and 2 μM CdCl_2_ for 7 days. (**C**) The proliferation rate of BMMSCs was detected by 5-ethynyl-2′-deoxyuridine (EdU). Scale bar, 100 μm. BMMSCs were exposed to 0, 0.1, or 0.2 μM CdCl_2_ for 24 h. (**D**) Cytoskeleton of BMMSCs were shown by immunofluorescence and confocal microscopic analyses Nuclei were stained blue with 4′,6-diamidino-2-phenylindole (DAPI), and cytoplasm was stained green and red with α-tubulin and F-actin. Scale bar, 20 μm.

**Figure 3 ijms-20-01519-f003:**
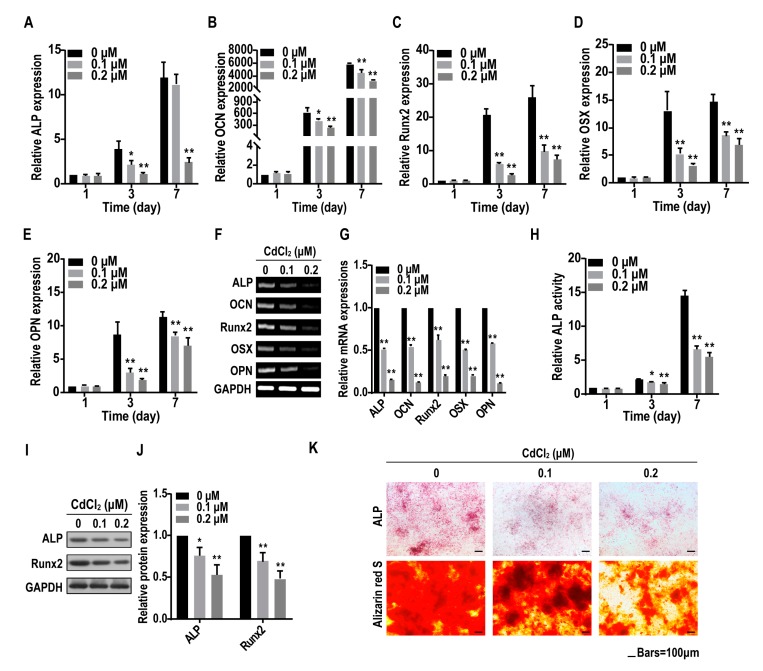
Cadmium inhibits the osteogenic differentiation of BMMSCs. BMMSCs were exposed to 0, 0.1, or 0.2 μM CdCl_2_ for 1, 3, or 7 days. (**A**–**E**) The mRNA levels of the osteogenic differentiation markers, such as alkaline phosphatase (ALP) (**A**), osteocalcin (*OCN*) (**B**), runt-related transcription factor 2 (*Runx2*), (**C**), osterix (*OSX*) (**D**), and osteopontin (*OPN*) (**E**) were measured by quantitative Reverse-Transcriptase Polymerase Chain Reaction (qRT-PCR) (mean ± SD, *n* = 3). * *p* < 0.05, ** *p* < 0.01, different from control BMMSCs. BMMSCs were exposed to 0, 0.1, or 0.2 μM CdCl_2_ for 7 days. (**F**) and (**G**) The mRNA levels of the osteogenic differentiation markers, *ALP*, *OCN*, *Runx2*, *OSX*, and *OPN* were determined by RT-PCR analysis (mean ± SD, *n* = 3). * *p* < 0.05, ** *p* < 0.01, different from control BMMSCs. BMMSCs were exposed to 0, 0.1, or 0.2 μM CdCl_2_ for 1, 3, or 7 days. (**H**) ALP activity was detected by ALP assay (mean ± SD, *n* = 3). * *p* < 0.05, ** *p* < 0.01, different from control BMMSCs. BMMSCs were exposed to 0, 0.1, or 0.2 μM CdCl_2_ for 7 days. (**I**) Western blots were performed, and (**J**) relative protein levels of ALP and Runx2 were determined (mean ± SD, *n* = 3). * *p* < 0.05, ** *p* < 0.01, different from control BMMSCs. BMMSCs were exposed to 0, 0.1, or 0.2 μM CdCl_2_ and subjected to osteogenic differentiation for 14 days. (**K**) The ALP content and the numbers of mineralization nodules were evaluated by ALP staining (upper) and alizarin red S staining (lower). Scale bar, 100 μm.

**Figure 4 ijms-20-01519-f004:**
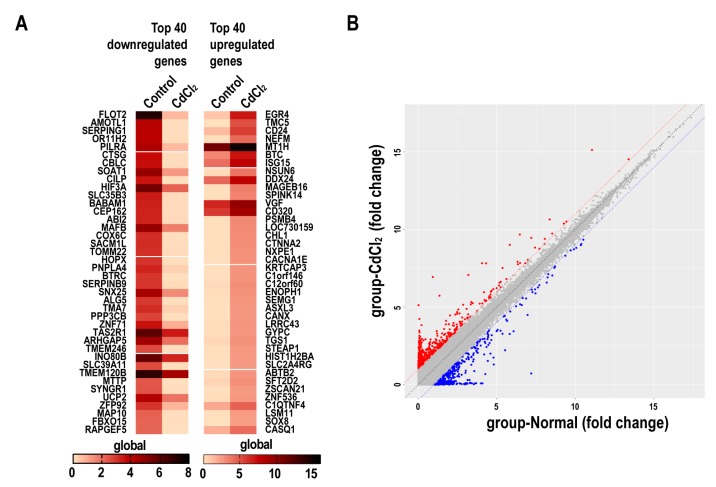
The effects of cadmium on mRNA expression profiles. BMMSCs were exposed to 0 or 0.2 μM CdCl_2_ for 24 h, and a mRNA microarray was accomplished with control BMMSCs and CdCl_2_-BMMSCs according to the standard protocols of Arrayster. (**A**) A heat map of the 40 down-regulated and 40 up-regulated mRNAs in control BMMSCs and CdCl_2_-BMMSCs. (**B**) The scatter plots were for variation of expression of mRNAs between control BMMSCs and Cd-BMMSCs.

**Figure 5 ijms-20-01519-f005:**
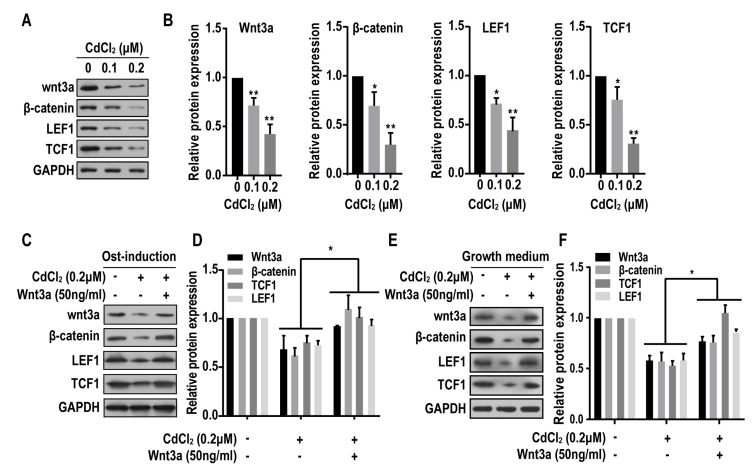
The effects of cadmium on Wnt/β-catenin pathway. BMMSCs were exposed to 0, 0.1 or 0.2 μM CdCl_2_ for 24 h. (**A**) Western blots were performed, and (**B**) relative protein levels of Wnt3a, β-catenin, LEF1 and TCF1 were determined (mean ± SD, *n* = 3). * *p* < 0.05, ** *p* < 0.01, different from control BMMSCs. BMMSCs were exposed to 0, 0.2 μM CdCl_2,_ or 50 ng/mL Wnt3a with the treatment of 0.2 μM CdCl_2_ in osteogenic medium for 24 h. (**C**) Western blots were performed, and (**D**) relative protein levels of Wnt3a, β-catenin, LEF1 and TCF1 were determined (mean ± SD, *n* = 3). * *p* < 0.05, different from BMMSCs treated with CdCl_2_. BMMSCs were exposed to 0, 0.2 μM CdCl_2_, or 50 ng/mL Wnt3a with the treatment of 0.2 μM CdCl_2_ in growth medium for 24 h. (**E**) Western blots were performed, and (**F**) relative protein levels of Wnt3a, β-catenin, LEF1 and TCF1 were determined (mean ± SD, *n* = 3). * *p* < 0.05, different from BMMSCs treated with CdCl_2_.

**Figure 6 ijms-20-01519-f006:**
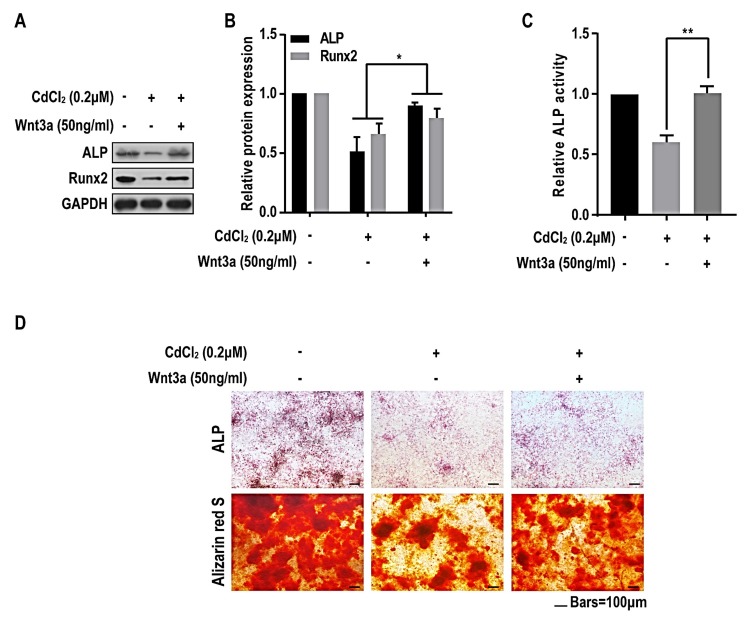
Wnt/β-catenin pathway is involved in cadmium-induced inhibition of osteoblast differentiation of BMMSCs. BMMSCs were exposed to 0, 0.2 μM CdCl_2_, or 50 ng/mL Wnt3a with the treatment of 0.2 μM CdCl_2_ in osteogenic medium for 7 days. (**A**) Western blots were performed, and (**B**) relative protein levels of ALP and Runx2 were determined (mean ± SD, *n* = 3). * *p* < 0.05, different from control BMMSCs. (**C**) ALP activity was detected by ALP assays (mean ± SD, *n* = 3). ** *p* < 0.01, different from control BMMSCs. BMMSCs were exposed to 0, 0.2 μM CdCl_2_, or 50 ng/mL Wnt3a with the treatment of 0.2 μM CdCl_2_ in osteogenic medium for 14 days. (**D**) The ALP content and the numbers of mineralization nodules were evaluated by ALP staining (upper) and alizarin red S staining (lower). Scale bar, 100 μm.

**Figure 7 ijms-20-01519-f007:**
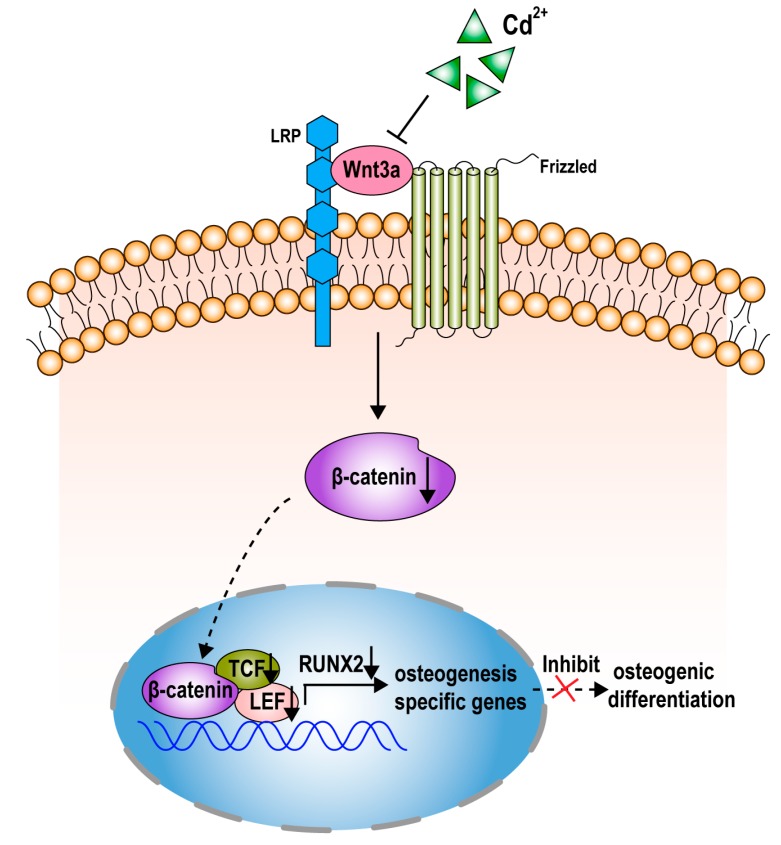
A schematic representation shows the proposed mechanism. The Wnt/β-catenin pathway is involved in the suppression of osteoblast differentiation of BMMSCs induced by CdCl_2_. A schematic representation of the proposed pathway for Wnt/β-catenin, which is inhibited by CdCl_2_, resulting in transcriptional reduction of Runx2 and the poor osteoblast differentiation of BMMSCs.

**Table 1 ijms-20-01519-t001:** Primer sequences used.

*GAPDH*	5′-GGCACAGTCAAGGCTGAGAATG-3′
	5′-ATGGTGGTGAAGACGCCAGTA-3′
*ALP*	5′-GGGACTGGTACTCGGACAAT-3′
	5′-GGCCTTCTCATCCAGTTCAT-3′
*OCN*	5′-CATGAGGACCCTCTCTCTGC-3′
	5′-TGGACATGAAGGCTTTGTCA-3′
*Runx2*	5′-GCACCCAGCCCATAATAGA-3′
	5′-TTGGAGCAAGGAGAACCC-3′
*OSX*	5′-AGCGACCACTTGAGCAAACAT-3′
	5′-GCGGCTGATTGGCTTCTTCT-3′
*OPN*	5′-CTTTCACTCCAATCGTCCCTAC-3′
	5′-CTGCCCTTTCCGTTGTTGTC-3′

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
