# Peer review of "Wnt/β-Catenin Pathway Is Involved in Cadmium-Induced Inhibition of Osteoblast Differentiation of Bone Marrow Mesenchymal Stem Cells"

_ijms, 2019, doi:10.3390/ijms20061519_

Round 1
Reviewer 1 Report
The manuscript by Qizhan Liu et al examines the underlying mechanism of low concentration of cadmium-mediated inhibition of osteogenic differentiation of bone marrow stromal cells. The detrimental effect of low-dose cadmium to osteogenic differentiation is attributed to dysfunction of canonical WNT signaling. The study is interesting but several crucial concerns need to be addressed before publication.
Major concern:
1. Considering the Tri-potent differentiation potential of BMSCs, chondrogenesis assay is also required in Fig 1.
2. In Fig 2C, the cytoskeleton of BMSCs was dramatically changed by low-dose cadmium, how did the authors make the conclusion that “These data indicated that low concentrations of CdCl2 had no effect on cell viability and cell morphology of BMMSCs” (Line 123).
Further, if the changed cytoskeleton is responsible for the impaired osteogenic differentiation potential? The authors need to explain, otherwise the data is out of place.
3. In addition to MTT assay, the proliferation rate of cadmium-treated BMSCs is also need to be assessed properly by EdU/BrdU assay and/or CFU-F assay.
4. Since the authors mentioned that cadmium resulted in epigenetic changes in the abstract, they need to show the relative data if there is any histone modification or DNA modification caused by cadmium treatment. They also need to determine whether the epigenetic changes also contribute to the phenotype or not.
5. For the osteogenic differentiation, the authors used classic osteogenic medium containing dexamethasone, β-glycerol phosphate, and ascorbic acid. For the rescue experiment, they added WNT3a into this system, and they concluded that CdCl2-induced inhibition of osteoblast differentiation is caused by WNT dysfunction. the question is that if canonical WNT signaling plays critical roles in this system? If the β-catenin level increased after BMSCs were treated with the osteogenic medium? To exclude the potential effects of the multi-factors in osteogenic medium, WNT-3a-treated ST2 cells (without osteogenic medium) will be more suitable for the mechanism study, which will narrow the differentiation system down to single factor.
Minor concern:
1. Overall the language is confusing and riddled with grammatical errors and typos. Having a native English speaker edit this manuscript would be helpful.
2. To make the manuscript readable, some figures need to be combined. (For example Fig 3 and Fig 4)
Author Response
March 18, 2019
Dr. Athena Wang
Assistant Editor, International Joural of Molecular Sciences
Manuscript ID: ijms-459205.
Original title: Wnt/β-catenin pathway is involved in low concentrations of cadmium-induced inhibition of osteoblast differentiation of bone marrow mesenchymal stem cells
Revised title: Wnt/β-catenin pathway is involved in cadmium-induced inhibition of osteoblast differentiation of bone marrow mesenchymal stem cells
Thank you for your e-mail dated March 4, 2019, regarding the review of our manuscript. We appreciate the comments from you and the reviewers. We have found the comments and suggestions helpful in preparation of the revised manuscript. We are glad to learn that you are willing to give further consideration to a suitably revised and modified version of the text.
In the revised manuscript, we have re-organized and re-written the unclear statements; provided additional details related to experimental methods in the Methods section; and invited Donald L. Hill (University of Alabama at Birmingham, USA), an experienced, English-speaking scientific editor, to edit and modify the text after we carefully checked the entire manuscript. Moreover, according to the suggestions of the reviewers, we have deleted “low concentrations of” in text and added some experiments. For example, we analyzed the cell proliferation by EdU assay, data of which were consistent with the MTT. Moreover, we also checked the effects of WNT-3a on levels of Wnt3a, β-catenin, LEF1, and TCF1 in no osteogenic medium-treated BMMSCs. In the discussion section, we have added more contents about the relationship between cytoskeletal changes and osteoblast differentiation.
We appreciate the general assessment of the manuscript by the reviewers and are grateful for their suggestions, which are helpful in improving of the quality of the present work and for our future studies. In the revised manuscript, all questions and suggestions from the reviewers have been addressed. Because the manuscript has been revised, the text and figure numbers have been changed. In order to save time for the reviewers, the tables below describe the changes.
Table 1. Changes in Text
Section | Pages | Changes |
Title | 1 | Deleted “low concentrations of”. |
Authors’ information | 1 | QQ emails were replaced. |
Abstract | 1 | Deleted “low concentrations of”. |
Abstract | 1 | Deleted epigenetic changes and. |
Introduction | 2 | Deleted “low concentrations of”. |
Introduction | 2 | Deleted “epigenetic” |
Results | 4 | Deleted “low concentrations of”. |
Results | 4 | Added information about the results of EdU. |
Results | 4 | Corrected mistakes. |
Results | 4 | Corrected unclear description of the results. |
Figure Legends | 5 | Added information about EdU. |
Figure Legends | 5 | Corrected mistakes. |
Results | 5 | Deleted “low concentrations of”. |
Results | 6 | Deleted “low concentrations of”. |
Figure Legends | 7 | Added information about RT-PCR, western blot, ALP activity, ALP staining and alizarin red S staining. |
Figure Legends | 8 | Deleted the original legends of figure 4. |
Results | 8 | Deleted “low concentrations of”. |
Results | 8 | Added information about the analysis of mRNA expression profiles. |
Results | 9 | Added information about Wnt signaling related mRNAs. |
Figure Legends | 10 | Deleted figure legends of figure 5 C and D. |
Results | 10 | Deleted “low concentrations of”. |
Results | 10 | Added results about the expression of Wnt3a, β-catenin, LEF1, and TCF1 in BMMSCs treated with CdCl2 (without osteogenic medium). |
Figure Legends | 11 | Deleted “low concentrations of”. |
Results | 11 | Deleted “low concentrations of”. |
Figure Legends | 13 | Deleted information about Wnt3a, β-catenin, LEF1, and TCF1 in BMMSCs treated with CdCl2. |
Discussion | 13 | Corrected mistakes. |
Discussion | 13 | Deleted “low concentrations of”. |
Discussion | 14 | Added information about the relationship between cytoskeletal changes and osteoblast differentiation. |
Discussion | 14 | Rewrite some overlap sentences. |
Discussion | 14 | Corrected mistakes. |
Discussion | 15 | Deleted “low concentrations of”. |
Discussion | 15 | Corrected mistakes. |
Discussion | 15 | Rewrite some overlap sentences. |
Methods | 15 | Added information about details of approval by a properly constituted research ethics committee. |
Methods | 16 | Added information about EdU assay methods. |
Methods | 18 | Added information about the antibody for β-catenin. |
Methods | 18 | Added information about samples and the analysis of heatmap, scatter and gene oncology methods. |
Conclusions | 19 | Deleted “low concentrations of”. |
Conclusions | 19 | Corrected mistakes. |
Conclusions | 19 | Added information about supplementary meterials. |
Table | 21 | Added a table about the detailed information of Wnt signaling related mRNAs in BMMSCs with CdCl2. |
Abbreviations | 21 | Added an abbreviation of 5-ethynyl-2’-deoxyuridine. |
Table 2. Changes in Figure Numbers and Changes in Figures (other than Figure Legends)
Number in Revised Manuscript | Number in Previous Manuscript |
Figure 2C | (New) Added EdU assay. |
Figure 2D | Figure 2C |
Figure 3F and 3G | Figure 4A and 4B. |
Figure 3H | Figures 3F. |
Figure 3I, 3J, and 3K | Figure 4C, 4D and 4E. |
Figures 4A and 4B | Figures 5A and 5B |
Figures 4B | Added values of x-axis and y-axis in the scatter plot. |
Figures 5A and 5B | Figures 5C and 5D |
Figure 5C, 5D | Figures 6A and 6B |
Figure 5E and 5F | (New) Added Western blots of Wnt3a, β-catenin, LEF1, and TCF1 in BMMSCs treated with CdCl2 (without osteogenic medium). |
Figure 6A, 6B, 6C and 6D | Figures 6C, 6D, 6E and 6F. |
The following, in order of the referees’ comments, are our responses:
Responses to Questions and Comments from Reviewer 1
The authors examine the underlying mechanism of low concentration of cadmium-mediated inhibition of osteogenic differentiation of bone marrow stromal cells. The detrimental effect of low-dose cadmium to osteogenic differentiation is attributed to dysfunction of canonical WNT signaling. The study is interesting but several crucial concerns need to be addressed.
Response: We appreciate the evaluation of the manuscript by the reviewer 1, and we are grateful for his/her suggestions. Based on the comments we received, careful modifications have been made. All questions and suggestions from the reviewer 1 have now been addressed. We trust that the revised manuscript will meet the reviewer’s standards. Below are our point-by-point responses to the reviewer’s comments and questions.
Major concern:
Q1: Considering the Tri-potent differentiation potential of BMSCs, chondrogenesis assay is also required in Fig 1.
Response: Thank you for your comments and valuable suggestions. According to the criteria to define BMMSCs, first, the used BMMSCs in our investigation were plastic-adherent when maintained in standard culture conditions. Second, ≥95% of BMMSCs had the expression of CD44 and CD90, while these cells lacked the expression of CD45; Third, these cells may differentiate to osteoblasts or adipocytes in vitro. Therefore, our BMMSCs from rats have potent differentiation potential.
Q2: In Fig 2C, the cytoskeleton of BMSCs was dramatically changed by low-dose cadmium, how did the authors make the conclusion that “These data indicated that low concentrations of CdCl2 had no effect on cell viability and cell morphology of BMMSCs” (Line 123)。
Further, if the changed cytoskeleton is responsible for the impaired osteogenic differentiation potential? The authors need to explain, otherwise the data is out of place.
Response: We apologize for the mistake description. According to the cytoskeleton of BMMSCs was dramatically changed by low-dose cadmium in Fig 2C, we have change the mistake description to “low concentrations of CdCl2 had no effect on cell viability, but damaged cell morphology of BMMSCs”.
In revised manuscript, we have added following explain: Actin cytoskeletal changes regulate the nuclear aggregation of the transcriptional coactivator with PDZ-binding motif (TAZ), which is indispensable for osteogenesis of bone mesenchymal stem cells (BMSCs) (Zhe et al. Simulated microgravity inhibits osteogenic differentiation of mesenchymal stem cells via depolymerizing F-actin to impede TAZ nuclear translocation. Sci Rep. 2016; 6:30322). Therefore, the changed cytoskeleton is responsible for the impaired osteogenic differentiation potential.
Q3: In addition to MTT assay, the proliferation rate of cadmium-treated BMSCs is also needed to be assessed properly by EdU/BrdU assay and/or CFU-F assay.
Response: As suggested by the reviewer, EdU assay was added to assess the proliferation rate of cadmium-treated BMMSCs in Figure 2, data of which were consistent with the results of MTT assay.
Q4: Since the authors mentioned that cadmium resulted in epigenetic changes in the abstract, they need to show the relative data if there is any histone modification or DNA modification caused by cadmium treatment. They also need to determine whether the epigenetic changes also contribute to the phenotype or not.
Response: We apologize for these unclear statements and “epigenetic changes” mentioned in our paper was a mistake in writing. We have deleted it.
Q5: For the osteogenic differentiation, the authors used classic osteogenic medium containing dexamethasone, β-glycerol phosphate, and ascorbic acid. For the rescue experiment, they added WNT3a into this system, and they concluded that CdCl2-induced inhibition of osteoblast differentiation is caused by WNT dysfunction. the question is that if canonical WNT signaling plays critical roles in this system? If the β-catenin level increased after BMSCs were treated with the osteogenic medium? To exclude the potential effects of the multi-factors in osteogenic medium, WNT-3a-treated ST2 cells (without osteogenic medium) will be more suitable for the mechanism study, which will narrow the differentiation system down to single factor.
Response: As suggested by the reviewer, we also checked the effects of WNT-3a on levels of Wnt3a, β-catenin, LEF1, and TCF1 in no osteogenic medium-treated BMMSCs. The results showed that the levels of Wnt3a, β-catenin, LEF1 and TCF1 were increased, which could exclude the potential effects of the multi-factors in osteogenic medium.
Minor concern:
Q1: Overall the language is confusing and riddled with grammatical errors and typos. Having a native English speaker edit this manuscript would be helpful.
Response: As suggested, we have carefully checked the manuscript and corrected grammar and spelling mistakes. In addition, to correct such mistakes, we have again invited Donald L. Hill (University of Alabama at Birmingham, USA), an experienced, English-speaking scientific editor, to edit and modify the text of revised manuscript after we carefully checked the entire revised manuscript.
Q2: To make the manuscript readable, some figures need to be combined. (For example Fig 3 and Fig 4)
Response: As suggested by the reviewer, we have combined original Figure 3 and Figure 4 into Figure 3 in revised manuscript. We have also changed original Figure 5 and Figure 6 into new Figure 4~6 according to our result description in revised manuscript.
We trust that the modifications made in the manuscript and the above responses meet your and the reviewers’ expectations. We look forward to hearing from you and to publication of this manuscript. Thank you for your consideration.
Sincerely,
Qizhan Liu, M.D., Ph.D.
Professor, Toxicology and Molecular Cell Biology
Reviewer 2 Report
Lu Wu and colleagues reported the effects of cadmium on osteoblast differentiation in bone marrow cells. They showed that cadmium inhibited expressions of osteoblast marker genes as well as calcium deposition of the cells. They conducted microarray analysis and proposed that Wnt/beta-catanin signaling was involved in the inhibitory effects of cadmium. Although the results in figures were clear, these data were not enough to support their description. There were several points to be addressed.
Major comment 1:
In lines 31, 39, 43, 232 and 434, the authors claimed that epigenetic changes were involved in the effects of cadmium on osteoblast differentiation. However, no result was shown in the manuscript. The authors need to revise the manuscript or show the results of the epigenetic change.
Major comment 2:
In lines 66 to 67, the authors claimed the advantage of in vitro cell culture in terms of labor, time and cost; however, they did not show any biological relevance of that cell culture to in vivo system. What concentration of cadmium was assumed to be exposed in the cells when people take cadmium in environmental pollutant. The authors wrote “0.1 μM and 0.2 μM" as low concentration of CdCl2”. Although these concentrations were low among concentrations that they tested, there was no rational to say “low”. The author need to revise the manuscript for this point. It also would be good to estimate the concentration of cadmium in our body in this context.
Major comment 3:
In figures 5C, 5D, 6A and 6B, the authors showed that cadmium inhibited protein expressions of Wnt3a, beta-catenin, LEF1 and TCF1. Were these mRNA expressions also decreased by cadmium in their microarray analysis. The authors need to show the data. In addition, which genes were actually listed as differentially expressed genes related to Wnt signaling in the microarray analysis. They need to show the gene list with value of the differentially expression.
Major comment 4:
In lines 244 to 246, the author claimed that BMMSCs from rat bone marrow and human bone marrow were used in the study. However, no results with human cells were shown in their data. The authors need to revise the manuscript or show the result.
Major comment 5:
In figure 7, the authors drawn the proposed mechanism of cadmium-mediated inhibitory action of osteoblast differentiation. In the figure, there is inhibitory arrow from Cd2+ to Wnt3a. Do the authors propose that Cd2+ inhibits physical interaction between Wnt3a and LRP or Frizzled. However, there was no evidence to support this drawing in the data. In addition, there was no description about the model in discussion. The authors need to revise the proposed model and discuss the model in discussion.
Minor comment 1:
The authors need to provide information about beta-catenin antibody in Materials and methods. Does the antibody recognize beta-catenin protein or specific phosphorylation site on it.
Minor comment 2:
The authors need to describe detail information about microarray analysis. How many biological replicates they had in the study. How did they make heatmap and scatter plot in figures 5A and 5B. What does color indicator in figure 5A mean. They need to add values of x-axis and y-axis in the scatter plot in figure 5B. What does red dots and blue dots indicate in the figure 5B. In lines 188 to 189, in the figure legend of figure 5B, the authors wrote that “BMMSCs were exposed to 0, 0.1 or 0.2 μM CdCl2 for 24h”. However, the scatter plot is two dimensions showing only two samples. In supplementary materials, what software did they use for gene ontology analysis. They need to revise these points.
Author Response
March 18, 2019
Dr. Athena Wang
Assistant Editor, International Joural of Molecular Sciences
Manuscript ID: ijms-459205.
Original title: Wnt/β-catenin pathway is involved in low concentrations of cadmium-induced inhibition of osteoblast differentiation of bone marrow mesenchymal stem cells
Revised title: Wnt/β-catenin pathway is involved in cadmium-induced inhibition of osteoblast differentiation of bone marrow mesenchymal stem cells
Thank you for your e-mail dated March 4, 2019, regarding the review of our manuscript. We appreciate the comments from you and the reviewers. We have found the comments and suggestions helpful in preparation of the revised manuscript. We are glad to learn that you are willing to give further consideration to a suitably revised and modified version of the text.
In the revised manuscript, we have re-organized and re-written the unclear statements; provided additional details related to experimental methods in the Methods section; and invited Donald L. Hill (University of Alabama at Birmingham, USA), an experienced, English-speaking scientific editor, to edit and modify the text after we carefully checked the entire manuscript. Moreover, according to the suggestions of the reviewers, we have deleted “low concentrations of” in text and added some experiments. For example, we analyzed the cell proliferation by EdU assay, data of which were consistent with the MTT. Moreover, we also checked the effects of WNT-3a on levels of Wnt3a, β-catenin, LEF1, and TCF1 in no osteogenic medium-treated BMMSCs. In the discussion section, we have added more contents about the relationship between cytoskeletal changes and osteoblast differentiation.
We appreciate the general assessment of the manuscript by the reviewers and are grateful for their suggestions, which are helpful in improving of the quality of the present work and for our future studies. In the revised manuscript, all questions and suggestions from the reviewers have been addressed. Because the manuscript has been revised, the text and figure numbers have been changed. In order to save time for the reviewers, the tables below describe the changes.
Table 1. Changes in Text
Section | Pages | Changes |
Title | 1 | Deleted “low concentrations of”. |
Authors’ information | 1 | QQ emails were replaced. |
Abstract | 1 | Deleted “low concentrations of”. |
Abstract | 1 | Deleted epigenetic changes and. |
Introduction | 2 | Deleted “low concentrations of”. |
Introduction | 2 | Deleted “epigenetic” |
Results | 4 | Deleted “low concentrations of”. |
Results | 4 | Added information about the results of EdU. |
Results | 4 | Corrected mistakes. |
Results | 4 | Corrected unclear description of the results. |
Figure Legends | 5 | Added information about EdU. |
Figure Legends | 5 | Corrected mistakes. |
Results | 5 | Deleted “low concentrations of”. |
Results | 6 | Deleted “low concentrations of”. |
Figure Legends | 7 | Added information about RT-PCR, western blot, ALP activity, ALP staining and alizarin red S staining. |
Figure Legends | 8 | Deleted the original legends of figure 4. |
Results | 8 | Deleted “low concentrations of”. |
Results | 8 | Added information about the analysis of mRNA expression profiles. |
Results | 9 | Added information about Wnt signaling related mRNAs. |
Figure Legends | 10 | Deleted figure legends of figure 5 C and D. |
Results | 10 | Deleted “low concentrations of”. |
Results | 10 | Added results about the expression of Wnt3a, β-catenin, LEF1, and TCF1 in BMMSCs treated with CdCl2 (without osteogenic medium). |
Figure Legends | 11 | Deleted “low concentrations of”. |
Results | 11 | Deleted “low concentrations of”. |
Figure Legends | 13 | Deleted information about Wnt3a, β-catenin, LEF1, and TCF1 in BMMSCs treated with CdCl2. |
Discussion | 13 | Corrected mistakes. |
Discussion | 13 | Deleted “low concentrations of”. |
Discussion | 14 | Added information about the relationship between cytoskeletal changes and osteoblast differentiation. |
Discussion | 14 | Rewrite some overlap sentences. |
Discussion | 14 | Corrected mistakes. |
Discussion | 15 | Deleted “low concentrations of”. |
Discussion | 15 | Corrected mistakes. |
Discussion | 15 | Rewrite some overlap sentences. |
Methods | 15 | Added information about details of approval by a properly constituted research ethics committee. |
Methods | 16 | Added information about EdU assay methods. |
Methods | 18 | Added information about the antibody for β-catenin. |
Methods | 18 | Added information about samples and the analysis of heatmap, scatter and gene oncology methods. |
Conclusions | 19 | Deleted “low concentrations of”. |
Conclusions | 19 | Corrected mistakes. |
Conclusions | 19 | Added information about supplementary meterials. |
Table | 21 | Added a table about the detailed information of Wnt signaling related mRNAs in BMMSCs with CdCl2. |
Abbreviations | 21 | Added an abbreviation of 5-ethynyl-2’-deoxyuridine. |
Table 2. Changes in Figure Numbers and Changes in Figures (other than Figure Legends)
Number in Revised Manuscript | Number in Previous Manuscript |
Figure 2C | (New) Added EdU assay. |
Figure 2D | Figure 2C |
Figure 3F and 3G | Figure 4A and 4B. |
Figure 3H | Figures 3F. |
Figure 3I, 3J, and 3K | Figure 4C, 4D and 4E. |
Figures 4A and 4B | Figures 5A and 5B |
Figures 4B | Added values of x-axis and y-axis in the scatter plot. |
Figures 5A and 5B | Figures 5C and 5D |
Figure 5C, 5D | Figures 6A and 6B |
Figure 5E and 5F | (New) Added Western blots of Wnt3a, β-catenin, LEF1, and TCF1 in BMMSCs treated with CdCl2 (without osteogenic medium). |
Figure 6A, 6B, 6C and 6D | Figures 6C, 6D, 6E and 6F. |
Responses to Questions and Comments from Reviewer 2
Lu Wu and colleagues reported the effects of cadmium on osteoblast differentiation in bone marrow cells. They showed that cadmium inhibited expressions of osteoblast marker genes as well as calcium deposition of the cells. They conducted microarray analysis and proposed that Wnt/beta-catanin signaling was involved in the inhibitory effects of cadmium. Although the results in figures were clear, these data were not enough to support their description. There were several points to be addressed.
Response: We appreciate the evaluation of the manuscript by the reviewer 2, and we are grateful for his/her suggestions. Based on the comments we received, careful modifications have been made. All questions and suggestions from the reviewer 2 have now been addressed. We trust that the revised manuscript will meet the reviewer’s standards. Below are our point-by-point responses to the reviewer’s comments and questions.
Major comments:
Q1: In lines 31, 39, 43, 232 and 434, the authors claimed that epigenetic changes were involved in the effects of cadmium on osteoblast differentiation. However, no result was shown in the manuscript. The authors need to revise the manuscript or show the results of the epigenetic change.
Response: We apologize for these mistaken statements and have deleted “epigenetic changes” mentioned in original manuscript.
Q2: In lines 66 to 67, the authors claimed the advantage of in vitro cell culture in terms of labor, time and cost; however, they did not show any biological relevance of that cell culture to in vivo system.
What concentration of cadmium was assumed to be exposed in the cells when people take cadmium in environmental pollutant. The authors wrote “0.1 μM and 0.2 μM" as low concentration of CdCl2”. Although these concentrations were low among concentrations that they tested, there was no rational to say “low”. The author need to revise the manuscript for this point. It also would be good to estimate the concentration of cadmium in our body in this context.
Response: We apologize for these unclear statements. According to reviewer’ suggestion, we have deleted that “the advantage of in vitro cell culture in terms of labor, time and cost” in revised manuscript.
Beside “low concentrations of cadmium have no effect on cell viability, and high concentrations of cadmium reduce cell viability of BMMSCs” in result description of Figure 2, we have changed all “low concentrations of cadmium” to “cadmium” including title in revised manuscript.
We appreciate reviewer’s suggestion about estimating the concentration of cadmium in our body. We have also the plan to detect cadmium levels of population exposed to environmental cadmium in our next investigation.
Q3: In figures 5C, 5D, 6A and 6B, the authors showed that cadmium inhibited protein expressions of Wnt3a, beta-catenin, LEF1 and TCF1. Were these mRNA expressions also decreased by cadmium in their microarray analysis. The authors need to show the data. In addition, which genes were actually listed as differentially expressed genes related to Wnt signaling in the microarray analysis. They need to show the gene list with value of the differentially expression.
Response: As suggested by the reviewer, we have added a list including Wnt3a, beta-catenin, LEF1, TCF1 and other differentially expressed genes related to Wnt signaling in the microarray analysis in table S2 of revised manuscript.
Q4: In lines 244 to 246, the author claimed that BMMSCs from rat bone marrow and human bone marrow were used in the study. However, no results with human cells were shown in their data. The authors need to revise the manuscript or show the result.
Response: We apologize for the mistaken description. We have deleted “human bone marrow” because we have only used BMMSCs from rat bone marrow.
Q5: In figure 7, the authors drawn the proposed mechanism of cadmium-mediated inhibitory action of osteoblast differentiation. In the figure, there is inhibitory arrow from Cd2+ to Wnt3a. Do the authors propose that Cd2+ inhibits physical interaction between Wnt3a and LRP or Frizzled. However, there was no evidence to support this drawing in the data. In addition, there was no description about the model in discussion. The authors need to revise the proposed model and discuss the model in discussion.
Response: To describe the relationship among Wnt3a and LRP or Frizzled, we have added following discussion in revised manuscript. The binding of Wnt with Frizzled initiates to form a protein complex consisting of Wnt, Frizzled, and LRP5/6, which subsequently activates activates intracellular disheveled (Dvl) to recruit the Axin complex, leaving free β-catenin in the cytosol (Yusuke et al. Wnt3a promotes differentiation of human bone marrow-derived mesenchymal stem cells into cementoblast-like cells. In Vitro Cell Dev Biol Anim. 2018;6). In our study, with the treatment of CdCl2, the expression of Wnt3a and β-catenin decreased, which means that less Wnt3a forms less the protein complex of Wnt, Frizzled, and LRP5/6.
Minor comments:
Q5: The authors need to provide information about beta-catenin antibody in Materials and methods. Does the antibody recognize beta-catenin protein or specific phosphorylation site on it?
Response: As suggested by the reviewer, we have added the detailed information about beta-catenin antibody in Materials and methods.
Q5: The authors need to describe detail information about microarray analysis. How many biological replicates they had in the study. How did they make heatmap and scatter plot in figures 5A and 5B. What does color indicator in figure 5A mean. They need to add values of x-axis and y-axis in the scatter plot in figure 5B. What does red dots and blue dots indicate in the figure 5B. In lines 188 to 189, in the figure legend of figure 5B, the authors wrote that “BMMSCs were exposed to 0, 0.1 or 0.2 μM CdCl2 for 24h”. However, the scatter plot is two dimensions showing only two samples. In supplementary materials, what software did they use for gene ontology analysis. They need to revise these points.
Response: As suggested by the reviewer, we have added detail information about microarray analysis. Specific representations of heatmap and scatter plot have been added to the materials and method (Page 19). In figure 4A, color indicator means gene expression, the darker the color, the more the gene expression. And in figure 5B, the red dots mean upregulated genes (> 2.0-fold change) and the blue dots mean downregulated genes (> 2.0-fold change). Related information about the analysis of gene oncology we have used had been added in the materials in our manuscript (Page 19).
Base on mRNAs differentially expressed between control (0 μM) and (0.2 μM) CdCl2-treated cells were identified through scatter plots, we chose further three concentrations (0, 0.1, 0.2 μM) of CdCl2 to investigate if the inhibition of osteoblast differentiation induced by CdCl2 in a concentration-dependent manner.
For gene ontology analysis, we have added that “The gene oncology was analyzed by bioinformatics using “ggplot2” package” in 4.15. Microarray for analysis of gene expression.
We trust that the modifications made in the manuscript and the above responses meet your and the reviewers’ expectations. We look forward to hearing from you and to publication of this manuscript. Thank you for your consideration.
Sincerely,
Qizhan Liu, M.D., Ph.D.
Professor, Toxicology and Molecular Cell Biology
Round 2
Reviewer 1 Report
All of my concerns have been addressed properly.
Reviewer 2 Report
The authors addressed all concerns raised and revised the manuscript properly.